# A Qualitative Study of Primary Care Physicians’ Experiences with Telemedicine during the COVID-19 Pandemic in North-Eastern Poland

**DOI:** 10.3390/ijerph20031963

**Published:** 2023-01-20

**Authors:** Karolina Pogorzelska, Ludmila Marcinowicz, Slawomir Chlabicz

**Affiliations:** 1Department of Family Medicine, Medical University of Bialystok, 15054 Bialystok, Poland; 2Department of Obstetrics, Gynecology and Maternity Care, Medical University of Bialystok, 15295 Bialystok, Poland

**Keywords:** COVID-19, qualitative research, Poland

## Abstract

(1) Background: Due to the COVID-19 pandemic, primary care clinics quickly moved to provide medical consultations via telemedicine, however, information about primary care professionals’ perspectives is limited. (2) Methods: Thirty semi-structured interviews with primary care professionals working in north-eastern Poland were conducted to assess their perspectives regarding the benefits and challenges of telemedicine. (3) Results: Primary care professionals highlighted that telemedicine increases access to medical services and reduces travel inconvenience. Remote consultation is not as time-consuming as in-person visits, which enables the provision of medical services to a greater number of patients which is particularly important in primary care. The inability to see patients and loss of non-verbal communication represent a significant difficulty in providing remote care. Primary care professionals indicated patients are not always able to express themselves sufficiently in a telephone call, which leads to performing medical consultations inefficiently. Physicians also pointed out that in particular medical cases, physical contact is still necessary to reach an accurate diagnosis and give the necessary treatment. Statements of the study participants also show that primary care professionals are satisfied with providing medical advice with telemedicine and show their interest in continuing remote consultation in the post-COVID era. (4) Conclusions: Primary care professionals have moved towards incorporating telemedicine into their daily routines due to the COVID-19 pandemic. Despite the many difficulties encountered, healthcare professionals have also noticed the benefits of telemedicine, especially during challenging circumstances. The study shows telemedicine to be a valuable tool in caring for patients, although it should be emphasized that face-to-face consultations cannot be fully replaced by remote consultations.

## 1. Introduction

The hastened growth of technology has led, among other things, to the development of telemedicine, i.e., medical care provided remotely to a patient in a separate location using computers, telephones, mobile phones, or the internet. The history of telemedicine dates back to the 1950s, when the Nebraska Psychiatric Institute and the Norfolk State Hospital established a closed-circuit television link between the two healthcare facilities for psychiatric consultations [1]. Over the next several years, telecommunication programs in highly developed countries began providing telehealth care between numerous smaller communities and larger medical facilities [2,3]. However, the usage of telemedicine was not homogeneous across various countries. This was due to differences in the awareness of the importance of telemedicine, variability in the quality of the infrastructures, and the level of informatics skills of healthcare professionals and patients [4,5,6]. Over time, telemedicine solutions were slowly developing to a small extent; nonetheless, their level was high in some countries, such as the USA, Canada, Australia, and Sweden [7,8,9,10].

While the number of telemedicine encounters all over the world has been increasing over the past few years, the outbreak of the COVID-19 pandemic substantially accelerated the implementation of telemedicine, broadly in healthcare institutions. In the COVID-19 pandemic, lockdowns, social distancing, and quarantine have been adopted as effective methods of reducing the spread of the SARS-CoV-2 virus [11]. Due to the stay-at-home measures put in place by governments during the COVID-19 pandemic, it was necessary to deliver medical care remotely to lessen the risk of direct human-to-human exposure [12]. Primary care, as the first level of contact of the population with the health care system, played a significant role in the COVID-19 response by providing medical care to patients with COVID-19 and ensuring continuity of care for non-COVID-19 patients [13].

Although telemedicine consultations with medical providers in Poland have been possible in accordance with the ordinance of the Ministry of Health since October 2019, as an alternative to face-to-face visits in primary care, they did not attract interest from healthcare professionals or patients [14]. The situation dramatically changed due to the outbreak of the COVID-19 pandemic after in March 2020 Poland’s government announced lockdown-type control measures. Therefore, primary care clinics quickly moved to provide the majority of consultations via telemedicine (mainly telephone) and in-person contact with a doctor was possible only after initial telephone consultation and upon meeting certain conditions. Due to public concerns about the excessive use of telemedicine, the Ministry of Health issued a legal act regulating indications for in-person visits [15]. During the COVID-19 pandemic, telemedicine was a subject of public and medical professionals growing concern with some of the public expressing dissatisfaction with limited access to face-to-face consultations. Primary care professionals faced the challenging tasks of providing medical care in previously unknown conditions.

Due to the limited number of studies on the experiences of primary care professionals, we sought to fill this gap by studying the perspectives of Polish healthcare professionals. To capture the experiences with the implementation of telemedicine during the COVID-19 pandemic, we conducted a qualitative study with primary care professionals.

The aim of the study was to enhance the understanding of how primary care healthcare professionals in Poland see the benefits and limitations of telemedicine, to evaluate their satisfaction with services delivered via telemedicine, and how they see the future use of telemedicine.

## 2. Materials and Methods

This study adopted a descriptive qualitative approach in order to obtain a rich and detailed account of the data [16]. The qualitative research was carried out using the interview technique, relevant to exploring primary care professionals’ experiences and opinions. Approval for conducting the study was granted by the Medical University of Bialystok (No. APK.002.146.2021).

### 2.1. Study Settings and Participants 

The research was conducted with primary health care professionals working in 8 primary care clinics in north-eastern Poland between August 2021 and April 2022. Thirty participants expressed interest in participating in the study. A total of 30 participants were included in the study (11 family medicine physicians, 13 family medicine resident physicians, 3 nurses, and 3 midwives). All participants were Polish.

### 2.2. Data Collection 

The semi-structured interviews with healthcare professionals were conducted over the telephone and based on the interview topic guide. The interview topic guide included the questions listed in Table 1 below. 

### 2.3. Interview Procedure

The interviews were performed at a time convenient for each participant. All the interviews were conducted in Polish by the same interviewer (paper contributor KP). The sampling of the study participants was designed to ensure maximum variability in relation to socio-demographic characteristics such as age, gender, and length of work experience [17]. Interviews were conducted until data saturation was reached. Interviews varied in length between 15 min and 44 min, with an average length of 27 min.

Data collection started with the interviews in August 2021 and ended in April 2022. 

Participation in the study was voluntary. Each primary care professional was asked for consent at the beginning of the interview, and oral informed consent was obtained from all participants. All interviews were audio recorded and then transcribed verbatim.

### 2.4. Data Analysis 

Thematic analysis was used in the process of identifying patterns or themes within qualitative data. The analysis process started with getting acquainted with the data. After reading the transcripts and looking at the data, an initial list of ideas was generated and seed codes were prepared. When all data were pre-coded, a list of potential topics was created. These topics were then reviewed critically. In the next phase, the topics and sub-topics were finally defined and named [18].

The analysis was conducted by two experts with experience in carrying out qualitative research. The process of coding and generating topics and subtopics was carried out by all members of the research team. 

### 2.5. Characteristics of Study Participants

Participant characteristics can be found in Table 2 below. The majority of participants (14) had 6–10 years of experience working as primary care professionals. The participants (*n* = 30) were between 28 and 59 years old (the average age was 41 years); there were 24 females and 6 males.

## 3. Results

The analysis of the content of the transcripts identified the following topics related to teleconsultations in primary care: (1) the advantages of telemedicine; (2) difficulties in implementing telemedicine; (3) ensuring the health safety of patients; (4) satisfaction with telemedicine; (5) prospective future use of telemedicine.

### 3.1. Advantages of Telemedicine

The participants of the study reported many benefits of telemedicine, including: (1) ensuring continuity of patient care; (2) preventing the spread of the SARS-CoV-2 virus; (3) increasing the availability of medical services; (4) time-saving.

#### 3.1.1. Ensuring Continuity of Patient Care

During the COVID-19 pandemic, telemedicine ensured continuity of patient care, both for those with SARS-CoV-2 and all other patients. 

“Let’s put it this way, telemedicine allowed us to continue treating patients”.(female, 58 years old, family medicine physician)

#### 3.1.2. Preventing the Spread of the SARS-CoV-2 Virus

An especially significant advantage of telemedicine was in the prevention of the spread of the SARS-CoV-2 virus. Both patients and healthcare professionals benefited from this. 

“Teleconsultations reduced the risk of staff contracting COVID-19 because many doctors had died because they got infected with COVID-19. I can say that teleconsultations saved many people’s lives by limiting infection and unnecessary contact between people”.(female, 58 years old, family medicine physician)

#### 3.1.3. Increasing the Availability of Medical Services

Study participants reported many examples showing that telemedicine increased access to medical services in various ways. On the one hand, a face-to-face medical consultation is not always necessary, and at the same time, it may be convenient for professionally active patients. 

“It was really convenient because many cases do not require a personal visit of the patient and a medical examination; taking history is often enough. It also really helps working or elderly people who have mobility problems”.(female, 31 years old, family medicine physician)

In addition, patients can seek medical advice during holidays or business trips. This form of benefit has remained the preferred choice of many patients.

“It seems to me that when teleconsultations are available, patients have more access to the doctor. They can call from anywhere. It often happens that patients who are away from their place of residence call and would not be able to get to the clinic because they are 200–300 km from the clinic, e.g., on a business trip, or have gone on vacation and their medications have run out. I think it has made medical advice more accessible to patients in general. There are a lot of patients who do not want to come to the clinic at all now. They want everything done over the phone”.(female, 36 years old, family medicine physician)

Telemedicine is convenient for people with disabilities or difficulties with arriving at the clinic. 

“It makes access to the doctor easier, especially in areas without public transport, in smaller towns where patients usually do not have a car and they are unable to reach the family doctor’s office, or due to disability”.(male, 32 years, family medicine resident)

#### 3.1.4. Time-Saving

Some participants indicated the possibility of performing more telemedicine consultations in comparison with face-to-face visits. 

”Patients who do not have to come in person take less time during teleconsultations”.(female, 31 years old, family medicine resident physician)

### 3.2. Difficulties in Telemedicine Implementation

The analysis of the content of the transcripts enabled the identification of difficulties on the part of the healthcare professional, difficulties on the part of the patient, as well as technical and organizational difficulties, both for the patient and the healthcare worker (Table 3).

#### 3.2.1. Difficulties on the Part of the Healthcare Professional Include the Inability to Perform a Physical Examination, Difficulties in Explaining Medical Recommendations to the Patient over the Phone, Inability to Communicate Non-Verbally, and Insufficient Knowledge of Healthcare Workers as to the Health Indications of Telemedicine

##### The Inability to Perform a Physical Examination

The main disadvantage of telemedicine is the inability to perform physical examinations. Physicians emphasized this point in complete agreement. 

”Of course, the big downside is the lack of physical examination of the patient”.(female, 33 years old, family medicine physician)

PCPs noted that complaints like chest pain, abdominal pain, and neurological symptoms are exceptionally challenging to assess without a physical examination.

##### Difficulties with Explaining Medical Recommendations to the Patient

Medical professionals also noticed that delivering medical recommendations to certain patients during telemedicine visits may be challenging. 

“I know that the patient forgets about 70% of instructions after leaving the doctor’s office, which is why we note them down on paper. Therefore, giving instructions on drug doses during a teleconsultation is more difficult”.(female, 58 years old, family medicine physician)

Nevertheless, physicians put effort into various ways of facilitating the patient’s understanding of the recommendations provided. 

“It was easier for me to instruct the patient during an eye-to-eye conversation rather than during teleconsultation. But I try to explain everything as clearly as possible to the patient, for example, I give them the phone number they should call next”.(female, 29 years old, family medicine resident physician)

##### The Lack of Non-Verbal Communication with the Patient 

Telemedicine visits can also introduce communication barriers caused by the loss of non-verbal communication, and the inability to notice and listen for nuances in the patient’s way of reacting to information and diagnosis via telephone. Physicians make diagnostic and therapeutic decisions based on information obtained from the patient.

“I have to rely only on what the patient says. I can only hear the messages, I can’t see the patient. I can’t see their body language, can I? How they respond to my questions”.(female, 56 years old, family medicine physician)

Physicians noted that physical proximity has great importance in the patient-physician relationship, and telemedicine cannot replace face-to-face consultations.

“Teleconsultations cannot completely replace in-person visits and I wouldn’t even want that. When I meet the patient, and this kind of relationship is created, this kind of contact is different. Knowing the patient is very important. If you know the patient, you build trust, you make associations, you understand the patient, it is really important. You can’t completely replace that with teleconsultations. That wouldn’t be good for anyone”.(male, 31 years old, family medicine physician)

##### Insufficient Knowledge of Healthcare Professionals about the Health Problems Appropriate for in Telemedicine

“Often a nurse can’t fully assess the patient’s condition and, for example, when making an appointment with us for a teleconsultation, when we have a full list of patients for a given day, we talk and notice that the patient requires an immediate in-person visit with a doctor, so this leads to inconveniences like we had to either cram these patients into our schedule or, alternatively, make arrangements with a colleague who will be in the afternoon”.(female, 33 years old, family medicine physician)

#### 3.2.2. Difficulty on the Part of the Patient

The following difficulties were found on the part of the patient: failure to prepare for telemedicine, difficulties in describing their health problems, naming ailments; contacting the physician on behalf of the patient; problems related to the patient’s age; and lack of knowledge (operating skills) of technical devices.

##### Patients Unprepared for Telemedicine Visits

Telemedicine was an entirely new form of medical service delivery that some patients had not experienced before. Many physicians observed that patients do not prepare adequately for their teleconsultation. Effective medical consultation is impossible to conduct due to the lack of information about the medical history or surgical history.

“Not every patient is prepared during teleconsultation, for example, they don’t have their medical records with them.” (female, 59 years old, family medicine physician)

It was noticed that many telemedicine consultations were taking place in an inappropriate setting which caused concerns about optimal patient comfort and privacy.

“Often, when the patient is not prepared for teleconsultation, because they are at work, there is no possibility of going outside and the visit is not high quality. Or when they are using public transport or doing shopping. For me as a doctor, and often for the patient too, these conditions don’t allow for a comfortable consultation”.(female, 31 years old, family medicine physician)

##### The Difficulty with Describing Medical Ailments Precisely

Family medicine physicians, based on their professional experience, know that the problem that the patient reports is not always the same problem that the doctor recognizes during a physical examination. Telemedicine prevents the physician from verifying the health problem that the patient is reporting.

“As physicians, we are keenly aware that often (especially with elderly patients) when they come to see a doctor, they report a health issue, but during the visit and physical examination, we actually notice an actual and significant health problem. Those are the patients who lose out during teleconsultations because they are unable to provide the exact information we expect”.(male, 32 years, family medicine physician)

Another physician observed additional obstacles to accessing telemedicine by elderly patients, especially those with cognitive dysfunction, dementia disorders, and deafness. 

“Senior patients who have poor hearing, patients who have problems such as dementia or with concentration, face-to-face contact is easier”.(female, 58 years old, family medicine physician)

Study participants notice the risk of committing a mistake during telemedicine due to difficulty with precisely describing medical ailments by patients.

“Not all patients can describe their symptoms over the phone. There are older patients who openly admit that they cannot, do not feel confident when talking on the phone, get nervous, and forget what they wanted to say. They need personal contact. The possibility of making an error is always greater compared with an in-person visit”.(female, 33 years old, family medicine physician)

##### Contacting the Physician on behalf of the Patient

Primary care professionals also observed that older patients often required substantial assistance in participating in a telemedicine visit e.g., due to their mental conditions. 

“Family members contacting us on behalf of the patient causes problems. A third person describes the way the patient is feeling. This is subjective and then we have to interpret it.”(female, 34 years old, family medicine resident physician)

##### Problems Related to the Lack of Knowledge (Operating Skills) of Technical Devices

Participants noticed that some patients may not have the skills necessary to benefit from telemedicine. 

“I think that most patients are able to undergo teleconsultation efficiently, but unfortunately, there are also some patients who don’t know how to benefit from them. They cannot use these medical services in the form of teleconsultation”.(male, 32 years, family medicine resident)

#### 3.2.3. Technical and Organization Difficulties

Some difficulties in the implementation of telemedicine affected both patients and healthcare professionals, such as legal restrictions on telemedicine, lack of mobile phone coverage, and insufficient technical equipment and telephone lines in family medicine clinics.

##### The Legal Restrictions on Telemedicine

During the COVID-19 pandemic, the implementation of telemedicine was regulated by relevant legal acts. Diagnostic and therapeutic procedures were imposed by the regulations of the Ministry of Health. In everyday practice, some legal provisions did not adjust properly, which complicated the work of family doctors. 

“I do not like these restrictions on teleconsultation of the elderly and children imposed by the Ministry of Health. I agree that, for example, children should be examined, but in the case of extending prescriptions, for example for milk formula, personal visits are not necessary. Much like with the elderly, it all depends on the situation. So such rigid regulations only make work more difficult and not easier, they cause additional stress. They are illogical, they make our work and the patient’s life more difficult”.(female, 56 years old, family medicine physician)

##### The Insufficient Number of Technical Equipment and Telephone Lines in the Family Medicine Clinic, the Lack of Mobile Phone Coverage

The implementation of telemedicine surprised not only patients but also employees and managers of family clinics. The problem was the insufficient number of telephone lines, the lack of mobile phone coverage, and insufficient technical equipment.

“The difficulties are mainly technical in nature and have forced many clinics to modernize. Most clinics had one contact number for registration, and one phone”.(male, 28 years old, family medicine resident physician)

### 3.3. Ensuring the Health Safety of Patients

The medical staff ensured the health safety of patients during telemedicine consultation through such activities as taking a thorough medical history, analyzing vital parameters, requesting additional tests, analyzing photos, engaging in face-to-face consultations, and protecting patients and employees against COVID-19.

#### 3.3.1. An In-depth Medical History

Study participants primarily focused on in-depth medical history. By talking with the patient carefully, first, they tried to gather information from the patient and then confirm that it was understood correctly. 

“First, I let the patient self-report their symptoms and, depending on what symptoms they reported, I asked about various things that could also occur in other similar diseases. I carefully asked and recorded whether the patient denied or confirmed the symptoms”.(female, 34 years old, family medicine resident physician)

#### 3.3.2. Analysis of Vital Parameters 

Patients were encouraged to measure vital parameters at home, such as body temperature, blood pressure, pulse, blood saturation, and glycaemia. The analysis of these parameters by the family doctor was an important element of telemedicine consultation. 

“Patients can measure blood glucose, blood pressure, pulse, body temperature and, for example, saturation measurements of COVID-19 patients are very important”.(female, 58 years old, family medicine physician)

#### 3.3.3. The Need for Additional Laboratory Tests

When telemedicine consultation was insufficient in the opinion of the physician and the patient’s health condition required a physical examination and additional tests, the patient was invited for a face-to-face consultation. In the case of bedridden patients, it was done by means of a home visit.

“A patient who needs an ECG, deeper diagnostics, X-ray, auscultation, we called them into the clinic, and in the case of bedridden patients, we decided to visit them at home”.(female, 57 years old, family medicine physician)

#### 3.3.4. Analysis of Patient’s Photo 

In particular cases, pictures of skin lesions sent to family medicine physicians may also provide additional information about the patient’s condition. 

“In my opinion, it is sometimes helpful to see pictures of skin changes, leg swelling, rashes and, for example, pictures showing the way the body has changed due to the disease”.(female, 56 years old, family medicine physician)

#### 3.3.5. Face-to-Face Consultation

Although in some situations the imposed legal act made the work of medical professionals more complicated, in other situations it facilitated decision-making and ensured the health safety of patients.

“When so-called alarm symptoms occur in the patient in the course of various disease entities. In addition, the situations that are specified in the regulations regarding teleconsultations, like the occurrence of ailments in young children, suspicion of cancer, or exacerbation of symptoms of chronic diseases”.(male, 33 years old, family medicine physician)

Moreover, the deterioration of the general condition of the patient and the recurrence of symptoms were indications for a face-to-face consultation at the family clinic. 

“An in-person visit should be made to patients suffering from COVID-19 who, after initial treatment, do not report improvement after a few days or patients who reported improvement and then there is a relapse of symptoms”.(female, 35 years old, family medicine resident physician)

There have also been health problems for children up to 6 years of age, for example, the lack of the expected weight gain of the newborn, difficulties in feeding the newborn, skin rashes, abnormally healing umbilical cord stump, and newborn jaundice. In such cases, there were medical indications for face-to-face consultations. 

“The mother reports that the baby does not latch on, when after weighing the parents noticed there was no expected weight gain, or when something disturbing happened to the wound, for example, after a cesarean section”.(female, 28 years old, midwife)

“I’d rather play it safe and go to the patient than have a child on my conscience because I didn’t notice severe jaundice. Unfortunately, this cannot be assessed during a teleconsultation, even online via a webcam”.(female, 33 years old, midwife)

#### 3.3.6. Protecting Patients and Workers from the SARS-CoV-2 Infection

Healthcare professionals have taken a number of steps at their family clinics to protect patients from the virus that causes COVID-19.

##### Alternating Personal Appointments and Telemedicine

Face-to-face consultations were arranged alternately with telemedicine in order to prevent patients from having contact with each other. 

“We try to make sure that teleconsultations take place between in-person visits so that patients do not contact each other in person, reducing the risk of coronavirus transmission”.(female, 30 years old, family medicine resident physician)

##### Separation of an Office for Performing Telemedicine and an Office for Face-to-Face Consultations

In addition, in primary care clinics, separate rooms for telemedicine visits and face-to-face consultations have been established, and telephone triage by nurses has been introduced.

“Initially, only teleconsultations took place, and then based on this teleconsultation a decision was made whether the patient should come for an in-person visit”.(female, 33 years old, family medicine resident physician)

“In a joint therapeutic team, meaning nurses, doctors, and midwives, we established certain rules that were written down. We bulleted, in a manner of speaking, what requires a personal visit according to the doctor’s opinion and what doesn’t. What can be, so to speak, arranged over the phone. We conduct such an initial interview at the registration desk; the patient calls, and then we know whether this situation can wait a few days or whether it needs to be attended to as soon as possible”.(female, 58 years old, nurse)

##### Purchase of the Necessary Equipment to Facilitate the Organization of Work in the Clinic

In some clinics, equipment was purchased to facilitate work during the COVID-19 pandemic, such as telephones, tablets, personal protective equipment, and ozonizers. 

“At our clinic, ozonation machines have been purchased and zones for infectious and non-infectious patients have been designated”.(female, 31 years old, family medicine physician)

### 3.4. Satisfaction with Telemedicine

Study participants reported that most of them are satisfied with providing telemedicine encounters remotely. As one physician pointed out:

“As a doctor, I am satisfied, because in fact I can do some things initially with the help of teleconsultation and fewer of these patients come to the clinic in person, the risk of exposure to COVID-19 for patients and staff is lower. I often order some tests, and I invite the patient for a personal visit with the result”.(female, 34 years old, family medicine physician)

“I mean, during the pandemic, I worked in 2 clinics, and in both of them, there was no issue in inviting a particular patient for an in-person visit after a teleconsultation. So in this regard, I didn’t feel any pressure or anxiety about taking good care of the patient”.(female, 35 years old, family medicine physician)

“Based on the experience of our clinic, teleconsultations have proven themselves. I don’t mean to give us straight A’s here. But it worked. It’s a really good method. Of course, we are talking about situations that, as I said at the beginning, can be, colloquially speaking, dealt with over the phone”.(female, 54 years old, nurse)

On the other hand, one physician admitted she did not find telemedicine attractive. 

“I mean, I don’t like these restrictions on the elderly and children imposed by the Ministry of Health. To be honest, I don’t like teleconsultations. In my opinion, they are suitable only for very specific situations, for example for providing the results of additional tests”. (female, 56 years old, family medicine physician)

### 3.5. Perspectives for the Future Use of Telemedicine

Commenting on telemedicine in primary care in general terms, study participants highlighted that this form of medical care will develop in the future, inter alia, because of a lack of human resources in healthcare. 

“These forms of telecare and telemedicine will certainly be developed because of staff shortages in every country in the world. All systems in the world are looking for solutions to replace doctors. The competencies of nurses and medical assistants are increasing, and the use of telemedicine enables the doctor to give out more advice”.(female, 58 years old, family medicine physician)

“Capabilities such as giving video advice are now available, and maybe some kind of an electronic stethoscope or electronic spirometer will be available, telemedicine will certainly develop in this direction, so I think that this is what progress in medicine is all about, and we will use such technical possibilities. I think it will also be possible to monitor the patient and help them in a way that they will not have to come to us at all. First of all, it is very convenient for the patient, but as always, everything will depend on the honesty of the doctor, on their knowledge. Only this kind of telemedicine is acceptable”.(female, 57 years old, family medicine physician)

Moreover, physicians underlined the importance of continuing medical care via telemedicine visits after the COVID-19 pandemic; for example: 

“Of course, the patient should still be able to contact the doctor remotely and, depending on the agreed upon solution, medical recommendations, and medical indications, whether an in-person visit is necessary or if the therapeutic and diagnostic process requires an in-person visit or a teleconsultation”.(male, 33 years old, family medicine resident physician)

## 4. Discussion

We used qualitative research methods to gain insight into the experiences of primary care professionals with telemedicine during the COVID-19 pandemic. 

The implementation of telemedicine in the early stages of the COVID-19 pandemic was essential to maintaining a continuity of medical care to patients and thus limiting the spread of the SARS-CoV-2 virus [19,20,21]. Despite the limited experience of telemedicine among healthcare professionals in Poland in the pre-pandemic era, our study revealed primary care professionals quickly adjusted to providing care remotely.

From primary care professionals’ perspective, the inability of seeing patients and the loss of non-verbal communication represent a significant difficulty in providing remote care. Those barriers are also similar to experiences reported by primary care physicians in different European countries [20,22,23,24]. Participants of our study noticed that communication and treatment through telemedicine could be more difficult compared to in-person appointments. The disruption of reading non-verbal components, such as body language, e.g., posture, gestures, and facial expressions, may lead to issues with understanding or establishing a proper physician-patient relationship. Patients are not always able to express themselves sufficiently in a telephone call, which in PCPs opinions leads to an inefficient telemedicine consultation [20,25,26]. Due to difficulties with naming medical ailments by patients, not infrequently physicians have to conduct remote consultations appropriately to obtain the necessary information. The results of a study by Agha et al. showed that during telemedicine encounters physicians tended to control the dialogue while patients were passive, which leads to an imbalance in physician-patient communication [27]. Those above-mentioned difficulties may force primary care professionals to take an in-depth medical history and familiarize themselves with various active listening techniques, such as paraphrasing and clarification to compensate for the lack of personal connection [24]. In line with cross-country comparison, in our qualitative study, PCPs emphasized that certain patient groups may not be able to benefit from telemedicine consultations, e.g., especially those with cognitive dysfunction, dementia disorders, and deafness [22,24]. Primary care physicians also pointed out that in particular medical cases physical contact is still necessary to make an accurate diagnosis and give the treatment needed [19]. What is more, PCPs emphasized that physical contact with the patient is crucial to building an effective patient-physician relationship, which is also reflected in the previous study [24]. Not infrequently, aspects such as obtaining a diagnosis, acquiring reciprocal reliance, and preparing the patients might be only achieved during a face-to-face visit, which can be the basis for planning and performing further telemedicine long-term follow-ups [28]. For remote follow-up, it is important to assess the right parameters that can be obtained remotely and provide the necessary information about the patient’s ability to use telemedicine [28]. Studies have shown telemedicine encounters are valuable and might be efficiently used for follow-ups of patients with: chronic pain [28] and chronic neurological disorders, e.g., Parkinson’s disease [29], after liver transplantations [30], diabetes, hypertension, [31] and rheumatoid arthritis [32].

Finally, this qualitative study found that PCPs highlighted many advantages of telemedicine despite some difficulties. Telemedicine increases access to medical services and reduces travel inconvenience [33,34,35,36,37]. Participants noticed remote consultation is not as time-consuming as in-person visits, which enables the provision of medical services to a greater number of patients which is particularly important in primary care. 

PCPs tried to provide the best care to patients despite various difficult circumstances arising due to the COVID-19 pandemic. Statements of the study participants also show that primary care professionals are satisfied with providing medical advice with telemedicine which is also reflected in a published study [38].

A hybrid model of telemedicine and in-person visits should be a standard in providing health services in primary health care. There is also substantial interest in continuing the use of virtual care among primary care practitioners post-pandemic, which was also underlined in a previously conducted study [38]. 

The outbreak of the COVID-19 pandemic has triggered the implementation of remote consultations and considerations on the legal aspects of telemedicine. The protection of medical data and patient privacy are among the most significant concerns in regards to telemedicine. The basis of ensuring the security of medical data is a secure computer system that allows access to medical data by authorized personnel. Fortunately, nowadays more attention is given to commercial services as well as healthcare facilities that collect and sell data for purposes unrelated to healthcare [39]. The widespread use of telemedicine raises additional threats to privacy and cybersecurity [39]. Due to the occurrence of cyberattacks in the past (e.g., WHO, the National Institutes of Health, and the Gates Foundation were presumably hacked in April 2020), telemedicine providers should alert their patients about potential risks associated with telehealth services [40]. Physicians who provide telemedicine should choose website services with the appropriate privacy policy, which protects the security and integrity of patient information [40]. Another aspect that needs to be carefully assessed is obtaining verbal informed consent which should be noted in the patient’s medical records [40]. Effective verification and confirmation of the patient’s identity is an essential part of ensuring patient privacy and preventing imposture. Patient identity should be confirmed by providing personal data during teleconsultation and based on data contained in medical records or showing any ID with the patient’s photo to the smartphone camera [41]. During teleconsultation, the patient should feel safe to openly discuss health problems in private, intimate conditions [41]. The above-mentioned good practice approaches allow for increasing patient’s safety during telemedicine encounters.

### Study Strengths and Limitations

A strong point of our study is that participants were recruited from different clinics and practice settings. All participants of the study were routinely conducting teleconsultations in their practice. Although we interviewed a limited number of primary care professionals, we reached theoretical saturation in our analyses. All primary care professionals provide medical care in north-eastern Poland, which may not represent the full picture of the situation in the entire country. The study was conducted during a global pandemic, which may not be reflective of non-pandemic behavior. 

## 5. Conclusions

In conclusion, primary care professionals have moved towards incorporating telemedicine into their daily routines due to the COVID-19 pandemic. Despite the many difficulties they encountered, healthcare professionals have also noticed the benefits of telemedicine, especially during challenging circumstances. The experience of primary care professionals shows the value of telemedicine with regard to caring for patients, although it should be emphasized that face-to-face consultations cannot be fully replaced by remote consultations. PCPs are satisfied with telemedicine and show their interest in continuing remote consultation in the post-COVID era. 

Primary care professionals’ experience with the adoption of telemedicine may help to develop standardized practice guidelines which could improve the future use of delivering medical care remotely in primary care.

## Figures and Tables

**Table 1 ijerph-20-01963-t001:** Questions included in the interview topic guide.

Guide Questions
Can you describe the way you care for patients in the context of the coronavirus epidemic?
How has the way healthcare services been provided changed?
How do you think teleconsultations affect the organization of work in primary health care? Do you experience any difficulties?
How do teleconsultations affect the availability of medical services?
What measures do you take to ensure the patient’s health safety during teleconsultation? When is a personal visit necessary?
Are teleconsultations a barrier to effective communication?
Are you satisfied with providing medical advice in teleconsultation form?
Should alternative methods of consultation be developed?

**Table 2 ijerph-20-01963-t002:** Characteristics of participants (*n* = 30).

Characteristics	Category	*N* (%)
Gender	Male	6 (20)
	Female	24 (80)
Primary Care Professional	Family Medicine Physician	11 (37)
	Family Medicine Resident	13 (43)
	Nurse	3 (10)
	Midwife	3 (10)
Age	<35	15 (50)
	35–50	5 (17)
	>50	10 (33)
Work experience	0–5	6 (20)
	6–10	14 (47)
	10+	10 (33)

**Table 3 ijerph-20-01963-t003:** Difficulties in providing telemedicine.

Topics	Sub-Topics
Difficulty on the part of the healthcare professional	Inability to perform physical examination
	Difficulties with explaining medical recommendations to the patient over the telephone
	Lack of non-verbal communication
	Insufficient knowledge of healthcare professionals about health related problems appropriate for telemedicine visits
Difficulty on the part of the patient	Patients unprepared for telemedicine visits
	Difficulty with describing medical ailments precisely
	Relatives contacting the physician on behalf of the patient
	Problems related to the lack of knowledge (operating skills) of technical devices
Technical and organizational difficulties	Legal restrictions on use of telemedicine
	The lack of mobile phone coverage, insufficient amount of technical equipment

## Data Availability

The datasets for this article cannot be made available by the authors given data protection rules.

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
