# Peer review of "A Qualitative Study of Primary Care Physicians’ Experiences with Telemedicine during the COVID-19 Pandemic in North-Eastern Poland"

_ijerph, 2023, doi:10.3390/ijerph20031963_

Round 1

Reviewer 1 Report

Thank you for the opportunity to review your manuscript.

Many similar articles were already published in the last two decades, also many of them were recently published with an aim to clarify the use of telemedicine during COVID-19 pandemics.

The title of your manuscript included "....COVID-19 pandemic in Poland", although in the part "Study strengths and limitations" you emphasized medical care in north-eastern Poland, which may not represent the full picture of the situation in the entire country (lines 500 and 501). Therefore, your title is a bit opposite to your aforementioned claim.

The concept of your manuscript is quite similar to recently published articles, although there is no table with description of the guide questions.

Overall, your manuscript did not offer new insights nor revealed any new features of the use of telemedicine which would be distinguishable from the other articles in scientific terms or its everyday use. 

Manuscript is not suitable for publishing in MDPI IJERPH.

Author Response

Dear Reviewer, 

Thank you for your time spent reading our paper and your suggestions.

In Poland, telemedicine is a relatively new healthcare delivery tool. The actual number of remote consultations between physicians and patients was rather low before the COVID-19 era. Due to COVID-19 pandemic telemedicine has been routinely offered as an alternative to face-to-face consultations in primary care in Poland. The vast majority of medical professionals and patients experienced teleconsultations for the first time. We wanted to capture the first insights into telemedicine utilization in healthcare setting.

The tittle of the article has been corrected. In addition, we prepared a table with description of the guide questions.

The improved changes are marked by red in reviewed manuscript.

Thank you.

Reviewer 2 Report

Thank you for submitting the manuscript. I have read your manuscript with great interest. The topic is very topical and in a constantly evolving field. Your paper addresses the topic in an elegant and precise manner. However, there are a couple of things I'd like you to implement.First you describe the great difficulty of examining the patient, that is, of carrying out the physical examination. This is certainly partly true, even if steps are being taken to overcome this limitation.You also address the legal issue of telemedicine, which is one of the thorniest points of telemedicine.

I would like you to elaborate on these two points, in particular on the indications for carrying out the televisit after the face-to-face visit (unless you have very advanced technological supports, and on the subject of privacy and data protection.Precisely for this reason, I suggest some references that will help you document yourself and that I would like you to mention: 

doi: 10.1007/s11606-017-4082-2.

DOI: 10.3390/ijerph182312416

  • DOI: 10.1016/S1474-4422(17)30167-9

doi: 10.5114/reum.2020.96630.

doi: 10.1016/j.ijmedinf.2020.104239.

I am convinced that with these few minor revisions your manuscript is worthy of publication.

Kind Regards

Author Response

Dear Reviewer,

Thank you for your time spent reading our paper and your suggestions.
We appreciate the comments received greatly, as they pointed out how to improve the manuscript. 

Information on data protection and patient privacy have been included in the discussion.
Additionally, we elaborated on the indications for performing teleconsultations after the face-to-face visits.
I would like to show appreciation for the reference suggestions, which we mentioned in our article.
The improved changes are marked by red in reviewed manuscript. 
Thank you.

Round 2

Reviewer 1 Report

To whom it may concern:

I appreciate the recent changes you made and your efforts to complete a study. As I stated initially, your study did not offer any new scientific or technical novelties in use of telemedicine. Therefore, I recommend  https://www.mdpi.com/journal/healthcare/sections/telehealth, especially  COVID-19: Digital Health Response around the World (Deadline: 28 February 2023)

Author Response

Dear Reviewer,

Thank you for your time spent reading our manuscript and we appreciate your review. 

Unfortunately, we cannot accept the suggestion of submitting the article to the aforementioned journal due to our university's accounting rules for scientific publications.